# Prediction of Specific Antibody- and Cell-Mediated Responses Using Baseline Immune Status Parameters of Individuals Received Measles–Mumps–Rubella Vaccine

**DOI:** 10.3390/v15020524

**Published:** 2023-02-13

**Authors:** Anna Toptygina, Dmitry Grebennikov, Gennady Bocharov

**Affiliations:** 1Gabrichevsky Research Institute for Epidemiology and Microbiology, 125212 Moscow, Russia; 2Marchuk Institute of Numerical Mathematics, Russian Academy of Sciences, (INM RAS), 119333 Moscow, Russia; 3Moscow Center for Fundamental and Applied Mathematics, INM RAS, 119333 Moscow, Russia; 4World-Class Research Center “Digital Biodesign and Personalized Healthcare”, Sechenov First Moscow State Medical University, 119991 Moscow, Russia; 5Institute of Computer Science and Mathematical Modelling, Sechenov First Moscow State Medical University, 119991 Moscow, Russia

**Keywords:** measles–mumps–rubella, post-vaccination immune response, correlation analysis, multivariable linear regression, predictors, antibody-mediated response, cell-mediated response

## Abstract

A successful vaccination implies the induction of effective specific immune responses. We intend to find biomarkers among various immune cell subpopulations, cytokines and antibodies that could be used to predict the levels of specific antibody- and cell-mediated responses after measles–mumps–rubella vaccination. We measured 59 baseline immune status parameters (frequencies of 42 immune cell subsets, levels of 13 cytokines, immunoglobulins) before vaccination and 13 response variables (specific IgA and IgG, antigen-induced IFN-γ production, CD107a expression on CD8${}^{+}$ T lymphocytes, and cellular proliferation levels by CFSE dilution) 6 weeks after vaccination for 19 individuals. Statistically significant Spearman correlations between some baseline parameters and response variables were found for each response variable (*p* < 0.05). Because of the low number of observations relative to the number of baseline parameters and missing data for some observations, we used three feature selection strategies to select potential predictors of the post-vaccination responses among baseline variables: (a) screening of the variables based on correlation analysis; (b) supervised screening based on the information of changes of baseline variables at day 7; and (c) implicit feature selection using regularization-based sparse regression. We identified optimal multivariate linear regression models for predicting the effectiveness of vaccination against measles–mumps–rubella using the baseline immune status parameters. It turned out that the sufficient number of predictor variables ranges from one to five, depending on the response variable of interest.

## 1. Introduction

At the beginning of the 21st century, the World Health Organization (WHO) announced the measles eradication program [1]. Despite tremendous effort to implement the plan, the targeted measles elimination was not achieved by 2015. The current objective is set to eliminate measles by 2025. However, in Europe, characterized by high vaccine coverage in the years 2017–2018, several measles resurgences took place, with several thousand cases reported [2,3,4]. Since the pre-vaccination era, it is known that natural measles infection induces lifelong protective immunity. It is considered that the measles vaccine also induces high protective immunity. However, the effectiveness and durability of the immunogenicity varies between vaccinated individuals. Besides humoral immunity, virus-specific cellular immunity influences the immunogenicity against measles. It has been documented that host genetics contributes to the inter-individual variability of immunogenicity, i.e., there are HLA allelic associations with either a high antibody-mediated immunity or a strong cellular immunity [5]. The SNPs in genes controlling cytokine production, e.g., the interferon gamma (IFN-γ), were revealed, which affect the mode and strength of measles-induced immunity [6]. Primary measles vaccine failure is reported for about 2–12% of children immunized at around one year of age [7]. In Russia, routine measles vaccination was started in 1967, and since 1986, the mandatory two-dose schedule of the measles vaccine has been followed. In many countries of the European and American regions, but not in all countries of the world, two-dose measles vaccination has also been introduced.

A multivariate analysis of the innate immune responses in humans following vaccination with yellow fever vaccine (YF17D) was used to examine the response predictors in [8]. The high-throughput data on gene expression profiling, multiplex analysis of cytokines and chemokines and flow cytometry in conjunction with computational models allowed the authors to identify gene signatures that correlate with and can be used to predict the magnitude of virus-specific CD8${}^{+}$ T cells and neutralizing antibody responses to vaccine [8]. In ref. [9], the immune status parameters were studied to predict the post-vaccination antibody responses to influenza vaccine. It was shown that the baseline PBMC subpopulation frequencies before vaccination suffice for constructing accurate predictive models. To this end, both correlation analysis and cross-validation-based predictive modeling were used. For training and validating predictive models, the subjects were randomly subdivided into training and testing sets (75% and 25%, respectively). The predictor parameters were iteratively selected based on the parameter’s strength of correlation with the end-point in the training set [9]. The antibody responses to meningococcus vaccine in healthy adults were studied using an integrative network modeling approach in [10]. Differently expressed genes together with the pathways (signaling and transcription) whose expression was correlated to antibody responses were identified. The mathematical modeling and analysis for linking data to models are based on the use of a broad range of mathematical statistics, information-theoretic, and machine learning algorithms [11,12,13]. In general, the development of a predictive model for personalized immune response is the major challenge in vaccinology [14]. A linear regression model linking the measures of vaccine-induced immune responses with various factors such as individual’s biomarkers was proposed, thus marking a first step towards a personalized predictive vaccinology.

Previously, we showed that the pre-vaccination host immune status, i.e., the peripheral blood lymphocyte subpopulation structure, the levels of major immunoglobulin classes and the serum cytokine profile, impact the height and durability of post-vaccination immunity [15]. In the current study, we examined the relationship between the subsets of peripheral blood lymphocytes and cytokines and the induced virus-specific cellular and humoral immune responses. The purpose of this study is to identify novel biomarker predictors of protective immunity following vaccination with measles–mumps–rubella vaccine.

The analysis of high-dimensional data in which the number of variables largely exceeds the small sample size is a challenge, leading to many problems in life sciences [16]. Various strategies for parameter selection can be used to identify the most informative ones. Several multivariate classification algorithms are explored to identify combinations of biomarkers that demonstrate sufficient predictive performance. In this study, we implemented three complementary approaches, i.e., correlation analysis, the supervised screening, and implicit feature selection using regularized (or penalized) regression.

In Section 2, the clinical data and the mathematical tools used for their analysis are described. In Section 3, we present the results of building the predictive models for vaccine-induced immune responses. The study is complemented by a discussion in Section 4.

## 2. Materials and Methods

### 2.1. Study Design

The study involved 19 children (9 boys and 10 girls) aged 1 to 2 years (average age 1 year 3 months). These children had not been vaccinated previously against measles, rubella and mumps, and did not have these diseases. All children were vaccinated with the Priorix vaccine (GlaxoSmithKline, Brentford, UK). Before vaccination, 1 week and 6 weeks after it, in total 6 mL of blood was taken from the cubital vein in test tubes Vacutaner with heparin, and gel to obtain serum. The blood samples were drawn immediately before the vaccine administration and on days 7 and 42 after vaccination. On days 0 and 7, we measured 59 immune status parameters, including the frequencies of 42 immune cell subsets, levels of 13 cytokines and antibody levels. On day 42, i.e., 6 weeks post vaccination, we measured the specific immune responses against each strain: specific IgA and IgG, Ag-induced IFN-γ production, CD107a expression on CD8${}^{+}$ T cells, and cellular proliferation levels by CFSE dilution (in total, 13 response variables). The overall details of the study design are summarized in Figure 1.

### 2.2. Isolation of Lymphocytes from Blood

Peripheral blood mononuclear cells (PBMCs) were isolated from heparinized peripheral blood samples by Ficoll-Histopaque density gradient centrifugation. Isolated PBMC were divided into 3 aliquots (see below).

### 2.3. Antibodies

Anti-CD3-FITC, anti-CD3-PerCP, anti-CD16/56-PE, anti-CD4-PerCP, anti-CD8-FITC, anti-CD8-PerCP, anti-CD38-PE, anti-DR-APC, anti-CD122-PE, anti-CD45RA-FITC, anti-CD45R0-PE, anti-CD62L-APC, anti-CD161-APC, anti-CD25-FITC, anti-CD127-PE, anti-CD27-FITC, anti-CD5-PerCP, anti-CD19-APC, CD107a-RE-Su5 (BD, Franklin Lakes, NJ, USA).

### 2.4. Immunofluorescent Staining

The first aliquot of PBMCs was incubated with anti-surface antigen antibodies at 4 ${}^{\circ }$C for 20 min, washed in phosphate buffered saline (PBS) and fixed with 1% formaldehyde in PBS. The specific cellular immune response to measles and rubella antigens was evaluated in 3 ways: Ag-specific cell detection, Ag-specific cell proliferation, and Ag-specific cytokine production. As antigens, a twin-ether extract of measles and rubella virus culture prepared by the E. Norrby [17] method was used. Antigens were preliminarily titrated on the mononuclear cells of several donors so that the selected concentration of antigen induced cellular responses in donors vaccinated and recovered from measles or rubella and did not cause such responses in donors who had not previously been in contact with these viruses.

### 2.5. Ag-Specific Cells Detection

The second aliquot of PBMCs was resuspended in 1 mL of RPMI-1640 medium supplemented with 10% calf fetal serum, 2 mM L-glutamine and gentamicin. Cell viability (according to trypan blue staining) was not lower than 95%. In three sterile conical tubes (volume 1.5 mL) were added: 50 μL of monensin solution (to a final concentration of 10 μM), a suspension of PBMCs ($10^{6}$ cells per tube), monoclonal antibodies to CD107a-PE-Cy5 (final dilution 1:100) and 10 μL of measles or rubella virus antigens; 10 μL of medium was added to the control tube instead of antigens. The same stock was used for all stimulations. The contents of each tube were mixed once with a pipette, and then tubes were centrifuged to precipitate cells (200 g, 1 min) and incubated for 15 h at 37 ${}^{\circ }$C in an atmosphere of 5% CO${}_{2}$ and 100% humidity. After incubation, the tubes were again centrifuged (500 g, 1 min). The supernatant was carefully taken, and the cells were resuspended in PBS, then the cells were stained with anti-CD8-FITC antibodies, washed in PBS and fixed with 1% formaldehyde in PBS.

### 2.6. Cell Proliferation

A third aliquot of PBMCs was resuspended in 1 mL of RPMI-1640 medium with gentamicin at a concentration of $10^{7}$ cells per mL. Cells were placed in a sterile round-bottom tube (2 mL volume) and 5- and 6-carboxyfluorescein diacetate-succinylmidyl ether (CFSE) was added at a concentration of 1 μg/mL. Lymphocytes were incubated at 37 ${}^{\circ }$C for 30 min. Then, the cells were washed 4 times with cold RPMI-1640, diluted with RPMI-1640 medium supplemented with 10% calf fetal serum, 2 mM L-glutamine and gentamicin to a concentration of $2\times 10^{6}$ cells and placed into the wells of a 4-well panel in a volume of 1 mL per well (Nunc). The first well was a negative control; in the second well, phytohemagglutinin (PHA) 5 μg/mL was added as a positive control; in the third well—measles antigens; in the fourth one—rubella antigens. Panels were incubated for 8 days at 37 ${}^{\circ }$C in an atmosphere of 5% CO${}_{2}$ and 100% humidity. At the end of the incubation, 200 μl of supernatants were taken from each well for analysis of the cytokine profile. Then, the cells were washed in PBS and fixed with 1% formaldehyde in PBS.

### 2.7. Flow Cytometry

All samples prepared for FACS analysis were analyzed within 24 h of staining on an FACSCalibur cytometer using CellQuest software (BD).

### 2.8. Detection of Cytokines

The levels of 13 cytokines—IL-1 (IL-1β), IL-2, IL-4, IL-5, IL-6, IL-7, IL-8, IL-10, IL-12, IL-17A, IFN-γ, TGF-β and TNF-α—were evaluated both in serum and in culture supernatants using a two-laser automated analyzer (Bio-plex Protein Assay System, Bio-Rad, Hercules, CA, USA) with commercial test systems (determined dynamic range 0.2–3200 pg/mL) in accordance with the manufacturer’s instructions. All reactions were performed in a 96-well plate format. The amount of cytokines in the test samples was determined using standard calibration dilutions, and cytokine concentrations were calculated automatically using the Bio-Plex Manager software.

### 2.9. Specific Immunoglobulin ELISA

Specific IgG antibodies to measles, rubella and mumps were determined in the serum samples by enzyme-linked immunosorbent assay (ELISA) (Euroimmun). Values of less than 0.01 IU/mL for measles antigens and less than 15 IU/mL for rubella and mumps antigens were considered negative. The amount of IgG antibodies considered as positive was greater than 0.25 IU/mL for measles viruses and 25 IU/mL for rubella and mumps viruses. To identify specific IgA antibodies to the studied viruses in the same test systems, the anti-IgG conjugate was replaced with an anti-IgA conjugate (clone 14A/1H9). The level of total antibodies in blood serum classes G, M, and A was determined by turbidimetry.

### 2.10. Data Source and Ethics Statement

The work was approved by the Ethics Committee of G.N. Gabrichevsky Research Institute for Epidemiology and Microbiology. Parents signed an informed consent for the participation of children in the research program.

### 2.11. Data Analysis

#### 2.11.1. Exploratory Analysis and Data Preprocessing

The baseline characteristics of the immune status of all subjects on day 0 (baseline) and day 7 post-vaccination are presented in Figure 2A and Figure 2B, respectively. The response variables are summarized in Figure 2C. All variables were tested for normality, the resulting D’Agostino’s K-squared omnibus tests didn’t show consistent normality among either variables or log-transformed variables (data not shown). Before training predictive models, we log-transformed the variables and then standardized design matrices and centered the responses (some models with log-transformed variables showed better performance). The missing data are indicated in Figure 2D; they were unavailable because of a technical issue rather than intentionally censored. The variables of baseline immune status TGF-β and IL-17A were excluded from the sets of potential predictors within feature selection strategies (see below) because of many missing values.

#### 2.11.2. Feature Selection Strategies for Building Predictive Models

We utilized three feature selection strategies to select potential predictors of the post-vaccination responses among baseline variables: (a) screening of the variables based on the correlation analysis; (b) supervised screening based on the information of changes of baseline variables at day 7; (c) implicit feature selection using regularization-based sparse regression (Figure 3). The following filtering criteria were used for screening using strategies (a) and (b). In strategy (a), the variables were selected with Spearman correlation *p*-value < 0.2; among them, up to 5 variables that correlate with each other as low as possible were selected. The candidate variables were also critically reviewed/approved taking into account their immunological function, which can be regarded as a curated purposeful selection strategy [18]. In strategy (b), the variables with significant and coherent changes on day 7 were selected in sub-strategy (b1) as determined by the paired *t*-test *p*-value < 0.05 and the Cohen’s d-value effect size > 1; the variables with insignificant changes on day 7 (which could be either due to a lack of change or because the changes among individuals are in different directions) were selected in sub-strategy (b2) as determined by the paired *t*-test *p*-value > 0.5 and the Cohen’s d-value effect size < 0.05.

#### 2.11.3. Correlation Analysis

For the analysis of pairwise baseline correlations and correlations between baseline variables and response variables, Spearman’s rank correlation coefficients were used as nonparametric measures of monotonic relationships. The *p*-values were obtained using Fisher transformation and Student’s t-distribution.

#### 2.11.4. Analysis of Changes from Baseline to Day 7 Postvaccination Data

For analysis of the effect size and statistical significance of variable changes from day 0 to day 7, we applied both the paired *t*-test and the Wilcoxon signed-rank test. The *p*-values were adjusted using Benjamini–Hochberg multiple testing correction procedure to control for the false discovery rate. The absolute value of Cohen’s d-values is reported as a measure of effect size for *t*-tests, the r-values ($\left|r\right|=\left|z\right|/\sqrt{n}$, where *z* is z-score) are reported for Wilcoxon tests.

#### 2.11.5. Sparse Regularization-Based Regression Models

Given the number of potential predictors p=59 is much larger than the number of observations with non-missing data n≤19, we applied regularization procedures for the linear regression models to handle the respective ill-posed problems. The value of regularization parameter determines the variable selection, i.e., the number of baseline parameters of immune status included in the model. There are various ways to regularize the linear regression, differing by the penalty function that is used to prevent the values of model coefficients from becoming too large (and thus screening out variables with zero coefficients). We used the minimax concave penalty (MCP) method implemented in R package ncvreg [19], which has two tuning parameters γ and λ (regularization parameters), and converges to popular lasso regression [20] with convex $l_{1}$-norm penalty as γ→∞. Contrary to the lasso regression, MCP-penalized regression is less biased because MCP applies less shrinkage to the large nonzero coefficients. Moreover, MCP-penalized regression is one of the sparse regression methods which have the oracle property, i.e., asymptotically, they recover the true variables and screen out the noise variables. The tuning parameters can be chosen using the cross-validation (CV) methods as those that provide the minimal out-of-sample prediction error estimate. The same out-of-sample error can be estimated using the information criteria (IC). We used AICc, the corrected for small sample sizes Akaike IC, which is a second-order estimate of the out-of-sample error. To determine the regularization parameters, we utilized the hybrid strategy combining the convexity diagnostics and AICc as described in [19]. For a given value of γ, we chose the value of λ using the AICc criterion and compared this value with the critical value $\lambda^{*}$, below which the objective function becomes locally nonconvex. If $\lambda <\lambda^{*}$, we increased γ to make the penalty more convex, and for $\lambda \gg \lambda^{*}$, we decreased γ to reduce the bias without fear of reaching the unstable region. This procedure was iterated for each model until the γ value reflected the balance between parsimony and convexity. The value of λ was chosen based on AICc. We also looked for the diagnostics for choosing the final value of λ for each model which are based on marginal false discovery rates (mFDR) of selected variables, calculated using the permutation of residuals [21]. The final value of λ was chosen to leave larger set of candidate variables to be used later for best-subset selection (see below).

#### 2.11.6. Choosing Final Predictive Models of Post-Vaccination Responses

For each set of candidate variables obtained using three feature selection strategies, multiple candidate linear regression models were built using all possible subsets of variables as independent variables of the model. These models were compared using repeated leave-group-out cross-validation by splitting the datasets into training and validation datasets randomly in approximately 80/20% proportions 1000 times without replacement. For each set of cross-validated models, the best model was selected by choosing a model with minimal average out-of-sample (computed on validation datasets) residuals, i.e., with minimal RMSE. The average out-of-sample coefficients of determination R${}^{2}$ of models are reported. The final predictive models for each post-vaccination response variable are chosen by comparing the final models obtained for each set of model candidates (Figure 3). The overall workflow for building predictive models of post-vaccination immune response is summarized in Figure 3.

## 3. Results

### 3.1. Correlation-Based Selection of Potential Predictor Variables

Screening the potential predictor variables using the correlation analysis between the vaccine-induced response and the baseline immune status parameters (Figure 4) resulted in the following selection:

For measles:**IgA:** CD127${}^{-}$CD4${}^{+}$CD25${}^{+}$, CD45RA${}^{+}$CD45R0${}^{+}$ and CD5${}^{+}$ lymphocyte frequencies, TNF-α, IL-17A.**IgG:** CD3${}^{+}$CD8${}^{+}$CD38${}^{+}$, CD5${}^{+}$, CD8${}^{+}$CD122${}^{+}$, CD3${}^{+}$CD16/56${}^{+}$ lymphocyte frequencies, IL-7, TGF-β, IL-1, IL-2, IL-12.**Proliferation:** CD3${}^{+}$CD16/56${}^{+}$, CD8${}^{+}$CD122${}^{+}$ lymphocyte frequencies, IL-7, IL-8, TGF-β.**CD8${}^{+}$CD107a${}^{+}$:** CD38${}^{+}$, CD3${}^{+}$CD8${}^{+}$CD122${}^{+}$ lymphocyte frequencies, IL-6, IL-8, IL-1, TNF-α.**IFN-γ:** CD3${}^{+}$CD8${}^{-}$CD122${}^{+}$, CD127${}^{+}$, CD19${}^{+}$CD27${}^{+}$ lymphocyte frequencies, IL-2, TGF-β.

For rubella:**IgA:** CD3${}^{+}$CD38${}^{+}$, CD25${}^{+}$CD127${}^{+}$, CD8${}^{+}$CD45R0${}^{+}$CD62L${}^{-}$, CD127${}^{-}$CD4${}^{+}$CD25${}^{+}$, CD27${}^{+}$ lymphocyte frequencies.**IgG:** CD45RA${}^{+}$, CD3${}^{-}$CD8${}^{-}$CD38${}^{+}$HLADR${}^{+}$, CD3${}^{+}$CD8${}^{-}$CD122${}^{+}$,CD4${}^{+}$CD45R0${}^{+}$CD62L${}^{-}$, CD19${}^{+}$CD27${}^{+}$ lymphocyte frequencies, IL-10, IFN-γ.**Proliferation:** CD8${}^{+}$CD122${}^{+}$, CD3${}^{+}$CD16/56${}^{+}$ lymphocyte frequencies, IL-7, IL-8, TGF-β.**CD8${}^{+}$CD107a${}^{+}$:** IL-8, IL-12, IL-6, CD3${}^{+}$CD8${}^{-}$CD122${}^{+}$, CD27${}^{+}$ lymphocyte frequencies.**IFN-γ:** CD3${}^{+}$38${}^{+}$, CD3${}^{+}$CD8${}^{-}$CD122${}^{+}$, CD4${}^{+}$CD45R0${}^{+}$CD62L${}^{+}$, CD27${}^{+}$CD4${}^{+}$CD45R0${}^{+}$CD62L${}^{-}$, lymphocyte frequencies.

For mumps:**IgA:** CD8${}^{+}$CD38${}^{+}$, CD3${}^{+}$CD122${}^{+}$, CD127${}^{+}$, CD8${}^{+}$CD45RA${}^{+}$CD62L${}^{-}$, CD19${}^{+}$CD27${}^{+}$ lymphocyte frequencies, IL-1.**IgG:** CD3${}^{+}$CD8${}^{-}$CD122${}^{+}$, CD19${}^{+}$CD27${}^{+}$, CD4${}^{+}$CD45R0${}^{+}$CD62L${}^{-}$, CD25${}^{+}$CD127${}^{+}$,CD4${}^{+}$CD45R0${}^{+}$CD62L${}^{+}$ lymphocyte frequencies, IL-2, IL-17A.

Additionally, the formally selected candidate predictor variables were critically reviewed/approved taking into account their immunological function.

### 3.2. Mode of Change-Based Selection of Potential Predictor Variables

Our second approach to selection of potential predictors of the response to vaccination variables is based on the analysis of changes from day 0 to day 7 of the baseline immune status parameters (Figure 5), i.e., the effect size and statistical significance. In the first strategy (b1), the variables with significant and coherent changes on day 7 were selected. These are (first 9 panels on Figure 5):CD3${}^{+}$HLADR${}^{+}$ lymphocyte frequency,CD3${}^{+}$CD8${}^{+}$CD38${}^{+}$ lymphocyte frequency,CD3${}^{+}$CD8${}^{-}$CD122${}^{+}$ lymphocyte frequency,CD3${}^{-}$CD8${}^{+}$CD122${}^{+}$ lymphocyte frequency,CD3${}^{-}$CD8${}^{-}$CD38${}^{+}$HLADR${}^{+}$ lymphocyte frequency,CD3${}^{+}$CD8${}^{+}$CD122${}^{+}$ lymphocyte frequency,CD3${}^{+}$CD45R0${}^{+}$CD4${}^{+}$CD161${}^{+}$ lymphocyte frequency,CD4${}^{+}$CD25${}^{+}$ lymphocyte frequency,IL-17A.

Following a second strategy (b2), the candidate variables with nonsignificant changes on day 7 (last 8 panels on Figure 5) were selected as follows:CD8${}^{+}$CD45R0${}^{+}$CD62L${}^{+}$ lymphocyte frequency,CD3${}^{-}$CD16/56${}^{+}$ lymphocyte frequency,IgM,CD3${}^{+}$ lymphocyte frequency,CD122${}^{+}$ lymphocyte frequency,IL-4,CD3${}^{+}$CD16/56${}^{+}$ lymphocyte frequency,CD45RA${}^{+}$ lymphocyte frequency.

### 3.3. Regularization-Based Selection of Predictor Variables

The third approach to the selection of the baseline predictor variables for the vaccine-induced immune responses was based on the application regularization procedure linked to the linear multivariate regression building and is implicit in its nature. The final identified set of the immune status parameters, which are the best predictors of the post-vaccination immune response variables, is presented in Table 1.

### 3.4. Predictive Models for Post-Vaccination Antibody and Cellular Responses

The general form of the multiple linear regression models reads
(1)$$Y=\beta_{0}+\sum_{i=1}^{p}\beta_{i}\times x_{i},$$
where *Y* stands for the response variable and $x_{i}$ are the predictor (or explanatory) variables. The identified predictors are presented in Table 1. The constants $\beta_{i}$ are called partial regression coefficients and $\beta_{0}$ is known as intercept.

During data pre-processing, the baseline variables $x_{i}$ are standardized, i.e., centered around their mean values $\mu_{i}$ and scaled by their standard deviations $\sigma_{i}$. We denote the standardization transformation as function *z*: z(x,μ,σ)=(x−μ)/σ. Before standardization, variables can be optionally logarithmically transformed (which is done if this transformation improves the model performance metrics). We denote the initial transformation as function g(x), which can be an identical function if logarithmic transformation is not applied ($g_{id}\left(x\right)=x$, $g_{id}^{-1}\left(x\right)=x$) or a function of natural logarithm ($g_{log}\left(x\right)=\log\left(x\right)$, $g_{log}^{-1}\left(x\right)=\exp\left(x\right)$). Thus, the final form of the predictive models reads
(2)$$Y=g^{-1}\left(\beta_{0}+\sum_{i=1}^{p}\beta_{i}\times z\left(g\left(x_{i}\right),\mu_{i},\sigma_{i}\right)\right).$$

The estimated values of the regression coefficients $\beta_{i}$, intercepts $\beta_{0}$ and scaling coefficients $\mu_{i}$, $\sigma_{i}$, as well as the form of function *g* are summarized in Table 2.

We have identified the set of multiple linear regression models for the vaccine-induced immune response to measles, rubella, and mumps. The corresponding predictor variables and coefficients are presented in Table 2. The number of predictors ranges from one to five. They appear to differ essentially between the response variables of interest, i.e., the antibody and cellular vaccine-induced immunity.

## 4. Discussion

Prediction of the vaccine-induced immune response using immune signatures measured at baseline prior to vaccination is a problem of fundamental and medical importance [16]. We presented a systematic analysis of the original data coming from study involving 19 children aged 1 to 2 years vaccinated against measles, rubella and mumps with the Priorix vaccine (GlaxoSmithKline). We have identified optimal multivariate linear regression models for predicting the effectiveness of vaccination against measles-mumps-rubella using the baseline immune status parameters. It turned out that the sufficient number of predictor variables ranges from one to five, depending on the response variable of interest. The results of our study provide a quantitative computational tool for predicting the effectiveness of vaccines on a personalized basis.

To identify baseline predictors of post-vaccination immune response, we utilized various screening strategies which are known as feature selection methods in machine learning literature [22]. The feature selection methods are used to construct competitive predictive models which are then compared to select the final optimal one. The correlation-based (screening strategy (a)) and regularization-based (implicit strategy (c)) methods do not imply assumptions about the nature of connection between predictive and response variables. In contrast, supervised learning strategy (b) assumes that baseline variables that either change (b1) or do not change (b2) coherently on day 7 among individuals are predictive of the immune response. This strategy produced the best final model for unspecific (b1) and measles-specific (b2) cell proliferation levels. The unspecific cell proliferation level is therefore predicted by the variables with high intersubject variation of day 7 to day 0 changes (coherent changes) while the specific one is predicted by the variables with high intrasubject variation (individual changes in different directions). Using various modeling approaches is required to ensure an overall robustness of the results.

We note that the results of predictive modeling should be regarded as part of explanatory analysis to select potential baseline predictors among a large subset of immune status parameters. Due to a limited number of observations for some baseline variables, they were excluded from being candidates for predictive modeling although some of them (TGF-β and IL-17A) correlate with response variables strongly (TGF-β correlates significantly and positively with measles-specific IgG, IL-17A—with measles-specific IgA). More research is needed to validate the results and refine the baseline predictors and their mechanistic links with various aspects of post-vaccination immune response.

Dimensionality reduction is a procedure required for the analysis of high-dimensional data collected for a small number of samples. It is recognized that an exhaustive exploration of high-dimensional data space is practically impossible [23]. One needs to explore various techniques to reduce the dimensionality. However, the building of low dimensional models which approximate high-dimensional data is based upon the assumption that a manifold with a small intrinsic dimension exists [24]. Our study suggests that this is likely to be the case in building low-dimensional multivariate linear regression models for predicting the vaccine-induced immunity from a plethora of baseline immune status parameters.

The calculated predictors for humoral and cellular responses to the three viruses included in the vaccine differ from each other. Indeed, the well-known fact that different people form humoral and cellular responses of different levels to the same dose of the vaccine is usually explained by the initial state of the immune system. But it is still not known for certain which parameters have such an effect. Having studied a wide range of parameters of the immune system before vaccination, we expected to get well-explained parameters as predictors from the point of view of modern ideas about the work of the immune system. The complexity of the modeling we have undertaken lies in the fact that the immune system functions as a network-type intertwined system, that is, various parameters can affect other parameters, and the effect may be pleiotropic. Having carried out mathematical modeling of the processes of formation of immune responses to vaccination, we tried to answer the question posed.

For humoral responses to the measles virus, the level of double-positive CD45RA${}^{+}$CD45RO${}^{+}$ cells and the concentration of TNF-α for IgA turned out to be significant, and for IgG—the concentration of IL-12 and IL-2 had strong positive effects, whereas the concentration of IL-1 had a negative one. In responses to the mumps virus, for IgA, the level of CD4${}^{+}$CD45RA${}^{+}$CD62L${}^{-}$ and the concentration of IL-1 turned out to be significant, and for IgG, the concentration of IL-2 had a positive effect, whereas the level of CD19${}^{+}$CD27${}^{+}$ had a negative one. In the humoral response to the rubella virus, the level of CD27${}^{+}$ and CD25${}^{+}$CD127${}^{+}$ cells showed a negative effect on IgA, and the level of CD45RA${}^{+}$ had a positive effect on the level of IgG while the levels of CD3${}^{+}$CD8${}^{-}$CD38${}^{+}$ and CD127${}^{+}$ had a negative effect, as did the concentrations of IFN-γ and IL-10. Why does one get such different predictors?

On the one hand, it should be noted that the composition of the used vaccine includes live attenuated viruses, and not a pure protein antigen. This is important because live viruses are able to evade various influences of the immune system in different ways, moreover, they can themselves affect the immune system. Different viruses use different ways of such influence. Therefore, a uniformity in immune responses to different viruses should not be expected. Also, the formation of IgG and IgA antibodies may differ. Whereas IgG antibodies are predominantly formed as a result of the maturation of B cells in the germinal centers, the IgA response can also develop along the extrafollicular pathway, and there are other cells, other interactions and other cytokines involved. On the other hand, the predictors we found do not contradict modern ideas about the functioning of the immune system. Indeed, the level of double positive CD45RA${}^{+}$CD45RO${}^{+}$ cells reflects the activation of lymphocytes, and TNF-α stimulates the immune response. Interestingly, the concentration of IL-2 had a positive effect on the level of IgG for measles and mumps viruses. This cytokine has a stimulatory effect on the immune response, and measles and mumps viruses are related viruses. The fact that the level of previously existing memory B cells (CD19${}^{+}$CD27${}^{+}$) has a negative impact on the formation of a new IgG response also does not contradict the known facts about competition between existing and newly formed memory cells. That the levels of T cells and activated T cells (CD27${}^{+}$ and CD25${}^{+}$CD127${}^{+}$) can negatively influence the level of the emerging IgA response to rubella is also not surprising, as is the negative impact of IL-10, IFN-γ and the level of initially activated helpers (CD3${}^{+}$CD8${}^{-}$CD38${}^{+}$).

With regard to cellular responses to measles and rubella viruses, it is very interesting that the level of antigen dependent lymphocyte proliferation in both cases was influenced by the level of NKT cells (CD3${}^{+}$CD16/56${}^{+}$). These cells are known to be involved in responses to non-protein antigens, but the effects of these cells are still not well understood. At the same time, the level of NK cells (CD3${}^{-}$CD16/56${}^{+}$) negatively affected the proliferative response to the measles virus. Probably, NK cells, being the first line of antiviral defense, can reduce the amount of the vaccine virus and thus reduce the proliferative response of the adaptive immune response to this virus. Interestingly, on the 7th day after vaccination, this population also continued to respond. It is quite natural that the level of CD8${}^{+}$CD107a${}^{+}$ in response to the measles virus is influenced by the level of activated cytotoxic cells (CD3${}^{+}$CD8${}^{+}$CD122${}^{+}$). The induction of IFN-γ production by this virus was negatively affected by the level of already activated helpers (CD3${}^{+}$CD8${}^{-}$CD122${}^{+}$), which is quite understandable, since if any immune response is already launched, it interferes with the formation of a new one. The level of CD8${}^{+}$CD107a${}^{+}$ responding to the rubella virus was affected by the level of CD27${}^{+}$ cells, this marker is present on all naive T cells. Also, CD27${}^{+}$ cells influenced the antigen-induced production of IFN-γ, which is not contradictory. In addition to all the above, it should be noted that the human population is very heterogeneous in terms of HLA, and this imposes restrictions on the presentation of certain antigens. This introduces additional interference into the process of modeling immune responses. Interestingly, the same immune parameters that responded on the 7th day after vaccination (CD3${}^{-}$CD8${}^{-}$CD38${}^{+}$HLADR${}^{+}$, CD3${}^{+}$HLADR${}^{+}$, CD3${}^{+}$CD8${}^{+}$CD122${}^{+}$, CD3${}^{+}$CD45R0${}^{+}$CD4${}^{+}$CD161${}^{+}$) turned out to be predictors for the level of the common proliferative response to PHA. These predictors were obtained by different methods, and such a coincidence, on the one hand, suggests that the response of the immune system to vaccination on the 7th day is due to the proliferative reaction of activated lymphocyte clones. On the other hand, such a coincidence indicates that the modeling-based choice of predictors is not random, but really reflects the processes under study.

Thus, it should be emphasized that the baseline immune status predictors identified by us do not contradict modern conceptions about development of the humoral and cellular immune response to viral antigens. We hope that further research in this area will clarify the range of predictors and deepen our understanding of the interaction of various parts of the immune system in the formation of immune responses to viruses.

Understanding the cellular and molecular mechanisms that control the ability of immune system to mount a protective immune response against various pathogens is a central problem in immunology. Modern research in immunology is characterized by an unprecedented level of detail that has progressed towards viewing the immune system as numerous components that function together as a whole network [25]. There are significant difficulties in analyzing the data being generated from high-throughput technologies for understanding immune system dynamics and functions. This calls for the application of mathematical modeling to complement the clinical studies with the aim to describe, analyze and predict the observable characteristics of infections [26]. Mathematical models describing the dynamics of virus infections is a rapidly developing theoretical area of mathematical immunology [27]. Mathematical models considering the spread of infections are broadly used in the evaluation of vaccination strategies at the population level [28,29]. However, practically relevant mathematical studies concerning the personalized predictive modeling of the immune responses to vaccines are still rather rare. The results of our study provide a basis for the development of the mechanistic models which could help to derive a better understanding of the mechanisms by which vaccines protect and with a targeted modulation of immune baseline parameters in order to improve vaccine outcomes [30].

Overall, systems approaches combining a high-dimensional characterization of the individual’s responses to antigenic perturbation with the information extraction power of computational modeling should pave the way to a rapid and transformative advances in modern systems vaccinology [31].

## Figures and Tables

**Figure 1 viruses-15-00524-f001:**
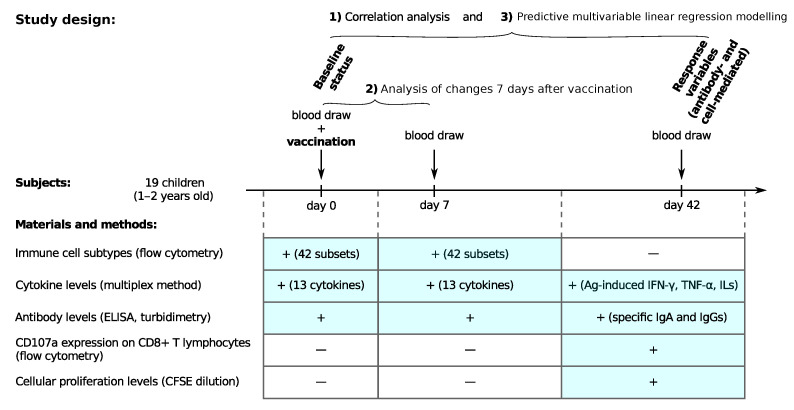
Study design and materials and methods.

**Figure 2 viruses-15-00524-f002:**
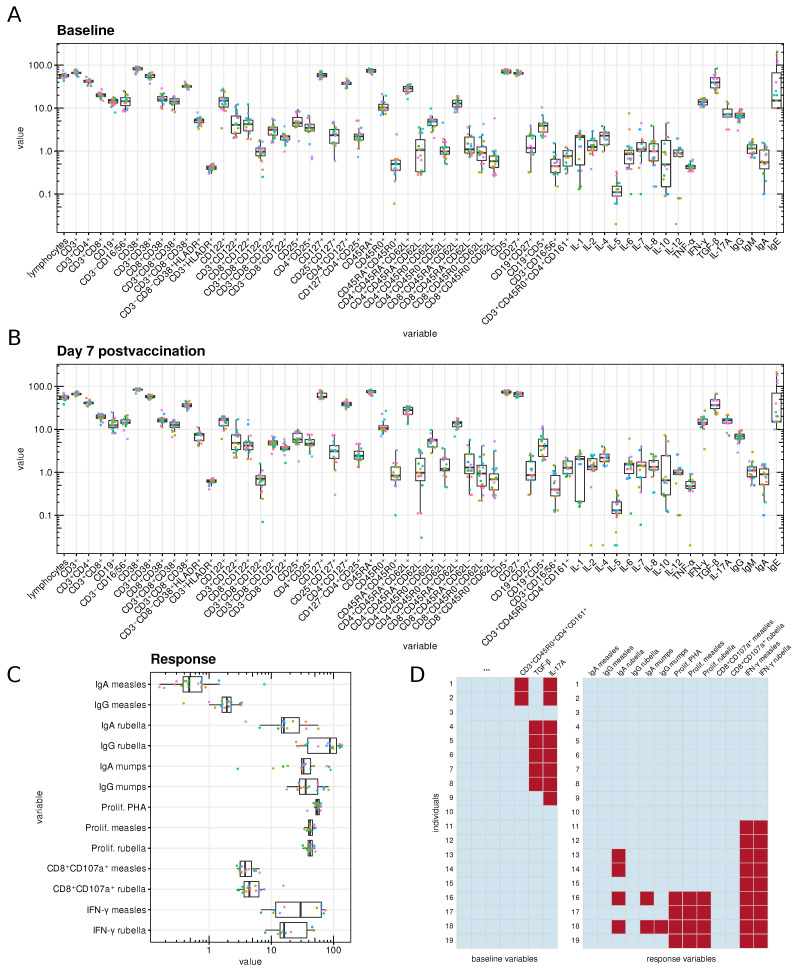
Collected data on baseline immune status and post-vaccination immune response. (**A**–**C**) Boxplots of measured immune status parameters at day 0 (**A**) and day 7 (**B**), and boxplot of post-vaccination response variables at day 42 (**C**). Points corresponding to the same individuals are marked with the same colors. (**D**) Indication of missing data for baseline and response variables (missing observations are colored in red).

**Figure 3 viruses-15-00524-f003:**
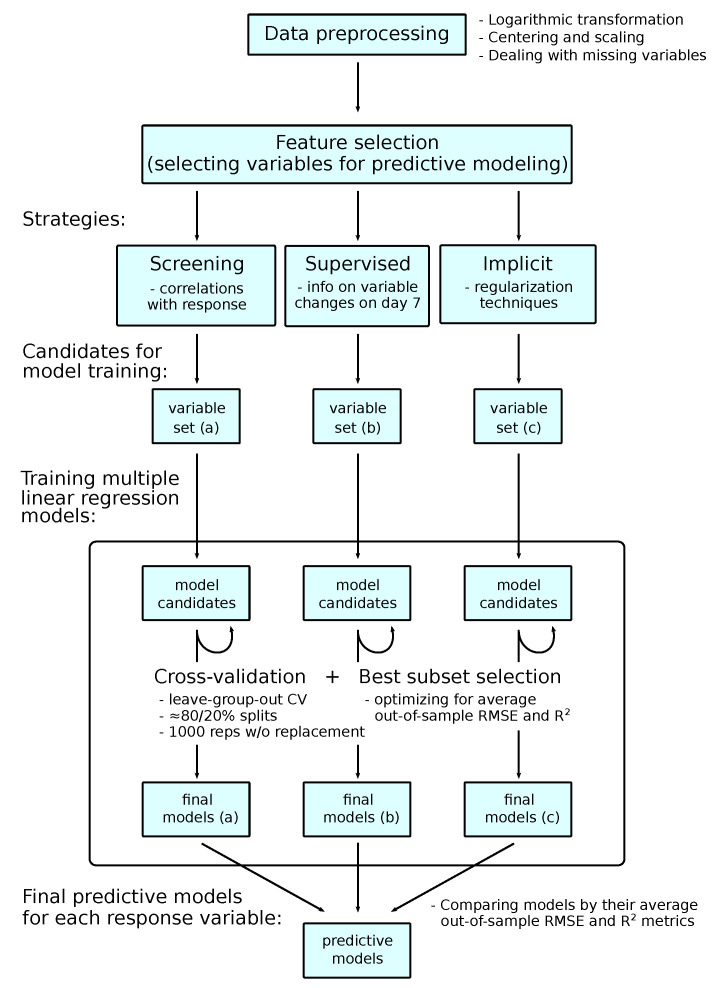
The workflow for building predictive models of post-vaccination immune response, which includes screening for potential predictors with three different screening strategies and comparing the candidate models using cross-validation.

**Figure 4 viruses-15-00524-f004:**
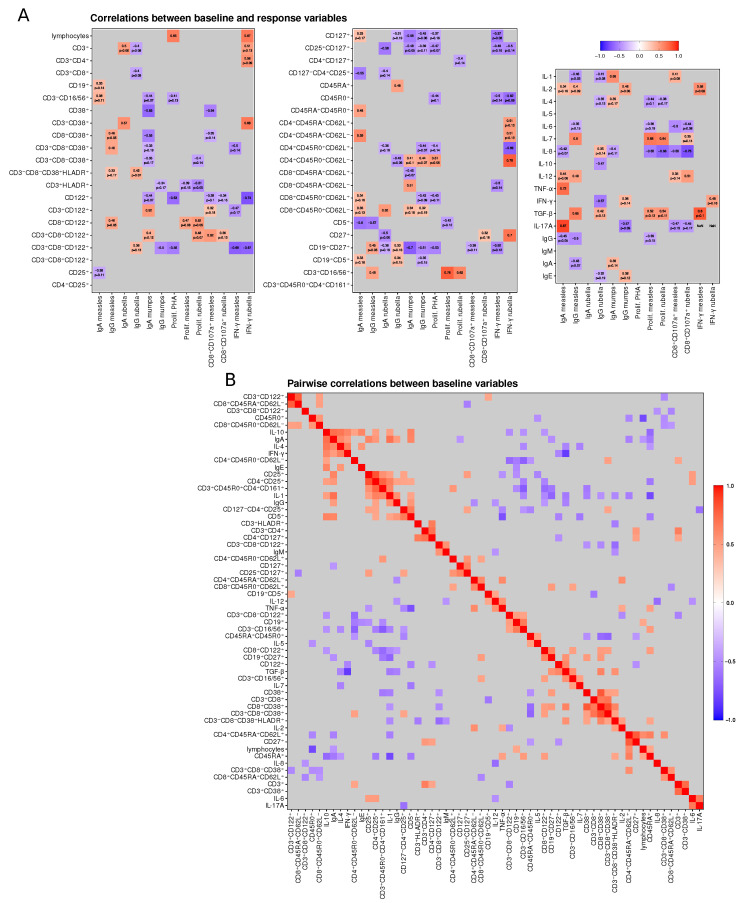
Correlation analysis. (**A**) Correlations between baseline immune status parameters and vaccine-specific immune response variables. Three levels of statistical significance are indicated: significant correlation coefficients (p<0.05) are shown, *p*-values are additionally written for correlations with 0.05≤p<0.2, and correlations with p≥0.2 are marked as gray cells. (**B**) Matrix of pairwise correlations for baseline immune status parameters sorted with hierarchical clustering ordering method. Correlations with significance level p≥0.05 are marked as gray cells.

**Figure 5 viruses-15-00524-f005:**
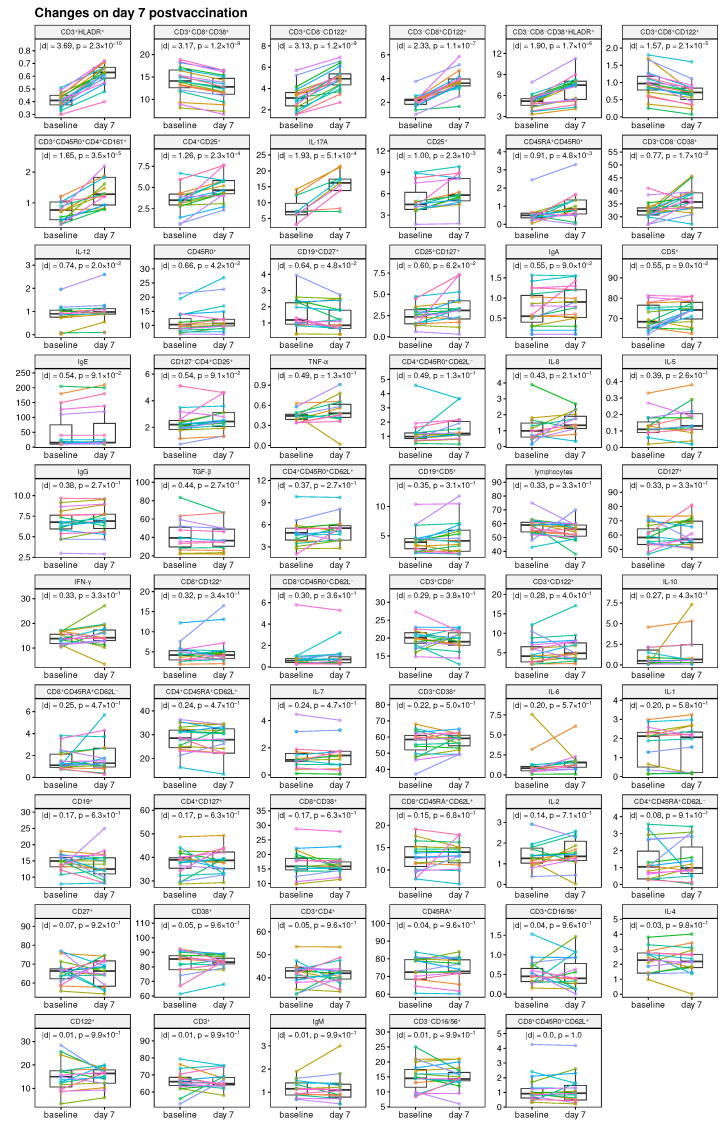
Analysis of the variable changes from baseline (day 0) to day 7 post-vaccination using the paired *t*-test. The effect size measured as the absolute value of the Cohen’s d-value and *p*-value adjusted to control over false discovery rate are indicated. The variables are sorted by the effect size. The first 9 variables were used as potential predictor candidates in strategy (b1), the last 8 variables—in strategy (b2).

**Table 1 viruses-15-00524-t001:** Summary of the final multilinear regression models predicting post-vaccination immune response variables: baseline predictor variables, cross-validation-based consistency metrics (mean out-of-sample error (RMSE), coefficient of determination ($R_{out}^{2}$)), model performance fitted on the complete dataset (adjusted $R_{adj}^{2}$, F-test *p*-value).

Response Variable *Y*	Baseline Predictor Variables $x_{i}$	Str.*	RMSE	$R_{\mathrm{out}}^{2}$	$R_{\mathrm{adj}}^{2}$	*p*-Value
**Measles:**						
IgA	TNF-α, CD45RA${}^{+}$CD45R0${}^{+}$	(a)	0.68	0.78	0.33	0.015
IgG	IL-12, IL-1, IL-2	(a)	0.6	0.75	0.36	0.02
Cell prolif.	CD3${}^{+}$CD16/56${}^{+}$, CD3${}^{-}$CD16/56${}^{+}$, CD3${}^{+}$, CD45RA${}^{+}$	(b2)	0.07	0.81	0.73	0.001
CD8${}^{+}$CD107a${}^{+}$	CD38${}^{+}$, CD3${}^{+}$CD8${}^{+}$CD122${}^{+}$	(a)	0.21	0.37	0.24	0.04
IFN-γ	CD3${}^{+}$CD8${}^{-}$CD122${}^{+}$, IL-2	(a)	14.2	1	0.52	0.03
**Mumps:**						
IgA	CD4${}^{+}$CD45RA${}^{+}$CD62L${}^{-}$, IL-1, CD27${}^{+}$	(c)	0.44	0.74	0.77	$6\;\cdot \;10^{-5}$
IgG	CD19${}^{+}$CD27${}^{+}$, IL-2	(a)	0.34	0.52	0.53	0.001
**Rubella:**						
IgA	CD27${}^{+}$, CD25${}^{+}$CD127${}^{+}$	(a)	10.41	0.81	0.56	0.003
IgG	CD45RA${}^{+}$, IFN-γ, CD127${}^{+}$, IL-10, CD3${}^{+}$CD8${}^{-}$CD38${}^{+}$	(c)	0.3	0.89	0.7	$6\;\cdot \;10^{-4}$
Cell prolif.	IL-7, CD3${}^{+}$CD16/56${}^{+}$	(a)	0.07	0.89	0.75	$9\;\cdot \;10^{-5}$
CD8${}^{+}$CD107a${}^{+}$	IL-8, CD27${}^{+}$	(a)	0.32	0.85	0.24	0.04
IFN-γ	CD27${}^{+}$	(a)	9.07	1	0.67	0.002
**PHA (unspecific):**						
Cell prolif.	CD3${}^{-}$CD8${}^{-}$CD38${}^{+}$HLADR${}^{+}$, CD3${}^{+}$HLADR${}^{+}$, CD3${}^{+}$CD8${}^{+}$CD122${}^{+}$, CD3${}^{+}$CD45R0${}^{+}$CD4${}^{+}$CD161${}^{+}$	(b1)	0.06	0.61	0.52	0.04

* Screening strategy that produced the best model: (a) correlation-based, (b) selecting variables that change on day 7 coherently (b1) or uncoherently (b2), (c) implicit regularization-based method.

**Table 2 viruses-15-00524-t002:** Summary of the parameters of the final multilinear regression models predicting post-vaccination immune response variables: regression coefficients $\beta_{i}$, intercepts $\beta_{0}$ and scaling coefficients $\mu_{i}$, $\sigma_{i}$, the form of transformation function *g*.

*Y*	*g*	$\beta_{0}$	$x_{i}$	$\beta_{i}$	$\mu_{i}$	$\sigma_{i}$
**Measles:**						
IgA	$g_{log}$	−0.52	TNF-α	0.45	−0.8365	0.1859
			CD45RA${}^{+}$CD45R0${}^{+}$	0.27	−0.8186	0.6961
IgG	$g_{id}$	2.03	IL-12	0.33	0.8679	0.5424
			IL-1	−0.32	1.5974	1.0041
			IL-2	0.12	1.4395	0.6621
Cell prolif.	$g_{log}$	3.75	CD3${}^{+}$CD16/56${}^{+}$	0.10	−0.7672	0.5367
			CD3${}^{-}$CD16/56${}^{+}$	−0.08	2.6545	0.3263
			CD3${}^{+}$	−0.05	4.1927	0.0950
			CD45RA${}^{+}$	0.04	4.2964	0.0895
CD8CD107a${}^{+}$	$g_{log}$	1.36	CD38${}^{+}$	−0.11	4.4108	0.1089
			CD3${}^{+}$CD8${}^{+}$CD122${}^{+}$	0.06	−0.1064	0.4953
IFN-γ	$g_{id}$	37	CD3${}^{+}$CD8${}^{-}$CD122${}^{+}$	−15	3.1568	1.1324
			IL-2	11	1.4395	0.6621
**Mumps:**						
IgA	$g_{log}$	3.45	IL-1	0.76	0.0789	1.1037
			CD4${}^{+}$CD45RA${}^{+}$CD62L${}^{-}$	0.35	−0.1043	0.9017
			CD27${}^{+}$	0.06	4.1852	0.0923
IgG	$g_{log}$	3.71	CD19${}^{+}$CD27${}^{+}$	−0.26	0.2228	0.6622
			IL-2	0.21	0.2634	0.4756
**Rubella:**						
IgA	$g_{id}$	20.4	CD27${}^{+}$	−8.3	65.975	6.1217
			CD25${}^{+}$CD127${}^{+}$	−6.9	2.5274	1.1903
IgG	$g_{log}$	4.24	CD45RA${}^{+}$	0.31	4.2964	0.0895
			IFN-γ	−0.24	2.6070	0.1597
			CD127${}^{+}$	−0.19	4.0726	0.1334
			CD3${}^{+}$CD8${}^{-}$CD38${}^{+}$	−0.16	3.4746	0.1004
			IL-10	−0.07	−0.6446	1.2909
Cell prolif.	$g_{log}$	3.72	IL-7	0.08	0.1041	0.8241
			CD3${}^{+}$CD16/56${}^{+}$	>0.05	−0.7672	0.5367
CD8${}^{+}$CD107a${}^{+}$	$g_{log}$	1.59	IL-8	−0.22	−0.1909	0.8052
			CD27${}^{+}$	0.06	4.1852	0.0923
IFN-γ	$g_{id}$	21	CD27${}^{+}$	13	65.975	6.1217
**PHA (unspecific):**						
Cell prolif.	$g_{log}$	4.04	CD3${}^{-}$CD8${}^{-}$CD38${}^{+}$HLADR${}^{+}$	−0.10	1.6134	0.1998
			CD3${}^{+}$HLADR${}^{+}$	−0.05	−0.8991	0.1372
			CD3${}^{+}$CD8${}^{+}$CD122${}^{+}$	0.04	−0.1064	0.4953
			CD3${}^{+}$CD45R0${}^{+}$CD4${}^{+}$CD161${}^{+}$	−0.01	−0.3762	0.4725

## Data Availability

The data presented in this study are available on request from the corresponding author.

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
