# Peer review of "Prediction of Specific Antibody- and Cell-Mediated Responses Using Baseline Immune Status Parameters of Individuals Received Measles–Mumps–Rubella Vaccine"

_viruses, 2023, doi:10.3390/v15020524_

Round 1
Reviewer 1 Report
The enclosed manuscript, “Prediction of specific antibody and cell-mediated responses using baseline immune status parameters of individuals received measles-mumps-rubella vaccine” by Toptygina and colleagues is a clinical modeling study that attempts to predict a set of immune response variables following characterization of individual participants. The results fit in with trends in the vaccinomics literature, including studies attempting to use systems biology techniques to discern and predict vaccine immunogenicity. The manuscript is interesting; however it could be improved by some deeper discussion of patient phenotypes and proposal of biological mechanisms. Recommend some corrections.
General:
There is overuse of the definite article “the”. The document should be checked for these translation errors.
Larger versions of the figures should be available, or layout should be changed in such a manner as the axis labels are visible at print size.
It would be better for the analyses code to be available in a Github for review.
This manuscript would benefit from an extensive supplement, especially considering that the figures are so complex.
Formatting/Grammar:
15: “linear”
21: Change “XXI” to “twenty-first”
33: change to “SNPs in genes controlling cytokine”
37,39: Two-time and double measles vaccine is better written as “two dose” or “two dose schedule”, or something like that
72: change “mots” to “most”
77-79: This paragraph can be deleted. It is implied that these things will be discussed.
85: Please list city for GSK production site or HQ
93: change “totally” to “in total”
103: Add city for BD
176: “Feature selection strategies” is more likely to appear in machine learning literature, and may not be clear to the readership of a virology journal. While it is correct terminology, it would be better not to complicate or alternatively, to provide extensive definitions.
330, 333: Sentence is repeated.
332: “from”
336: The
343: “linear”
Findings:
22: Cite WHO documentation for the elimination goals, and specify WHO regions in the statement.
29: Provide citations for variability of immunogenicity, since this is the rationale for the study.
31: The Poland study (citation 5) does not support variability of protection, only variability of immunogenicity.
154: Provide authoritative citation for these protective values or otherwise justify their selection.
174: It would be beneficial to more thoroughly discuss the censored data points. Was this technical error or was there an issue with the subjects?
186: It appears that there are multiple tests being applied to select features in strategy B. Why is it necessary, if the paired T test showed lack of significance, to continue on with using those variables in b2?
251,290, 314: While it was mentioned in methods that lymphocyte subset frequency is used as a feature variable here, the use of frequency measurement should be explicitly mentioned in the results section when discussing the predictors, instead of just the name of the subset. The reporting gives the impression that any detectable presence of those subsets (instead of their frequencies) is the actual reported predictor. Even better, the lists of predictors should be displayed in a table.
339: It would be beneficial to discuss immunological mechanisms that are accessible from these selected predictor sets, and to align those findings with other vaccinomics literature that has suggested these types of predictors.
Author Response
Response to Reviewer 1 comments
The enclosed manuscript, “Prediction of specific antibody and cell-mediated responses using baseline immune status parameters of individuals received measles-mumps-rubella vaccine” by Toptygina and colleagues is a clinical modeling study that attempts to predict a set of immune response variables following characterization of individual participants. The results fit in with trends in the vaccinomics literature, including studies attempting to use systems biology techniques to discern and predict vaccine immunogenicity. The manuscript is interesting; however it could be improved by some deeper discussion of patient phenotypes and proposal of biological mechanisms. Recommend some corrections.
Our response:
We thank the Reviewer for insightful comments and the thorough work on our manuscript. All the recommendations have been addressed in the revised manuscript as described below.
General:
(1) There is overuse of the definite article “the”. The document should be checked for these translation errors.
Our response:
The document has been additionally checked for the errors.
(2) Larger versions of the figures should be available, or layout should be changed in such a manner as the axis labels are visible at print size.
Our response:
Larger versions of Figures 1, 3 and 4 have been provided.
(3) It would be better for the analyses code to be available in a Github for review.
Our response:
Our analysis of the data is based on a number of algorithms implemented in R, rather than a single code. We would be happy to provide an advice upon request.
(4) This manuscript would benefit from an extensive supplement, especially considering that the figures are so complex.
Our response:
The raw clinical data presented in Figure 2 and the analysis results displayed in enlarged Figures 4-5 are described in detail in Sections 2 and 3, respectively. We could provide as a supplement the Excel file with raw data if needed (please, note that the data are statistically summarized in Figure 2).
Formatting/Grammar:
(5) 15: “linear”
Our response:
Corrected.
(6) 21: Change “XXI” to “twenty-first”
Our response:
Corrected.
(7) 33: change to “SNPs in genes controlling cytokine”
Our response:
Corrected.
(8) 37,39: Two-time and double measles vaccine is better written as “two dose” or “two dose schedule”, or something like that
Our response:
Corrected.
(9) 72: change “mots” to “most”
Our response:
Corrected.
(10) 77-79: This paragraph can be deleted. It is implied that these things will be discussed.
Our response:
Actually, it is quite common to clearly outline an overall structure of the research manuscripts at the end of the Introduction section. If we may, we would prefer to keep the paragraph.
(11) 85: Please list city for GSK production site or HQ
Our response:
Provided: ‘Brentford GB’.
(12) 93: change “totally” to “in total”
Our response:
Corrected.
(13) 103: Add city for BD
Our response:
Provided: ‘Franklin Lakes, New Jersey, USA’.
(14) 176: “Feature selection strategies” is more likely to appear in machine learning literature, and may not be clear to the readership of a virology journal. While it is correct terminology, it would be better not to complicate or alternatively, to provide extensive definitions.
Our response:
We have provided a paragraph with necessary definitions in Discussion section, as follows:
“To identify baseline predictors of post-vaccination immune response, we utilized various screening strategies which are known as feature selection methods in machine learning literature [1]. The feature selection methods are used to construct competitive predictive models which are then compared to select the final optimal one. The correlation-based (screening strategy (a)) and regularization-based (implicit strategy (c)) methods do not imply assumptions about the nature of connection between predictive and response variables. In contrast, supervised learning strategy (b) assumes that baseline variables that either change (b1) or do not change (b2) coherently on day 7 among individuals are predictive of the immune response. This strategy produced the best final model for unspecific and measles-specific cell proliferation levels. The unspecific response is predicted with parameters that changed coherently across individuals while the measles-specific cell proliferation levels are explained with variables which change individually in different directions during the first 7 days post-vaccination.
We note that the results of predictive modeling should be regarded as part of explanatory analysis to select potential baseline predictors among a large subset of immune status parameters. Due to a limited number of observations for some baseline variables, they were excluded from being candidates for predictive modeling although some of them (TGF-β and IL-17A) correlate with response variables strongly (TGF-β correlates significantly and positively with measles-specific IgG, IL-17A – with measles-specific IgA). More research is needed to validate the results and refine the baseline predictors and their mechanistic links with various aspects of post-vaccination immune response.”
[1] Kuhn, M.; Johnson, K. Applied Predictive Modeling; Springer New York: New York, NY, 2013; ISBN 9781461468486.
(15) 330, 333: Sentence is repeated.
Our response:
The repetition has been removed.
(16) 332: “from”
Our response:
Corrected.
(17) 336: The
Our response:
Not clear to us.
(18) 343: “linear”
Our response:
Corrected.
Findings:
(19) 22: Cite WHO documentation for the elimination goals, and specify WHO regions in the statement.
Our response:
The respective source, i.e. “World Health Organization (WHO). Eliminating measles and rubella and preventing congenital rubella infection. WHO European Region strategic plan 2005–2010 / World Health Organization 2012. – 44 p.” available at http://www.euro.who.int/__data/assets/pdf_file/0008/79028/E87772.pdf has been added to the references list, please see [4].
(20) 29: Provide citations for variability of immunogenicity, since this is the rationale for the study.
Our response:
Actually, we refer to the Poland’s team studies cited as [6,7] in the revised version.
(21) 31: The Poland study (citation 5) does not support variability of protection, only variability of immunogenicity.
Our response:
We have replaced „variability of protection“ to „variability of immunogenicity“.
(22) 154: Provide authoritative citation for these protective values or otherwise justify their selection.
Our response:
We have changed „protective“ to „positive“.
(23) 174: It would be beneficial to more thoroughly discuss the censored data points. Was this technical error or was there an issue with the subjects?
Our response:
The mentioned missing data were not censored. It was just a technical issue of sampling.
(24) 186: It appears that there are multiple tests being applied to select features in strategy B. Why is it necessary, if the paired T test showed lack of significance, to continue on with using those variables in b2?
Our response:
We have provided a commnet on this in the Discussion section. The rationale is provided by the supervised learning strategy which assumes that baseline variables that either change (b1) or do not change (b2) coherently on day 7 among individuals are predictive of the immune response. This strategy produced the best final model for unspecific (b1) and measles-specific (b2) cell proliferation levels. The specific cell proliferation level is therefore predicted by the variables with high intersubject variation of day 7 to day 0 changes (coherent changes) while the unspecific one is predicted by the variables with high intrasubject variation (individual changes in different directions).
(25) 251,290, 314: While it was mentioned in methods that lymphocyte subset frequency is used as a feature variable here, the use of frequency measurement should be explicitly mentioned in the results section when discussing the predictors, instead of just the name of the subset. The reporting gives the impression that any detectable presence of those subsets (instead of their frequencies) is the actual reported predictor. Even better, the lists of predictors should be displayed in a table.
Our response:
In the revised version, we have explicitly indicated that lymphocyte subset frequency are used.
In our view, the lists of predictors identified by various methods are clearly presented in subsections 3.1-3.3.
(26) 339: It would be beneficial to discuss immunological mechanisms that are accessible from these selected predictor sets, and to align those findings with other vaccinomics literature that has suggested these types of predictors.
Our response:
We have provided an extensive discussion of the implications concerning the immunological mechanisms underlying the identified predictors of responses to the vaccine:
“The calculated predictors for humoral and cellular responses to the three viruses included in the vaccine differ from each other. Indeed, the well-known fact that different people form humoral and cellular responses of different levels to the same dose of the vaccine is usually explained by the initial state of the immune system. But it is still not known for certain which parameters have such an effect. Having studied a wide range of parameters of the immune system before vaccination, we expected to get well-explained parameters as predictors from the point of view of modern ideas about the work of the immune system. The complexity of the modeling we have undertaken lies in the fact that the immune system functions as a network intertwined system, that is, various parameters can affect other parameters, and the effect may be pleiotropic. Having carried out mathematical modeling of the processes of formation of immune responses to vaccination, we tried to answer the question posed.
So for humoral responses to the measles virus, the level of double-positive CD45RA+CD45RO+ cells and the concentration of TNF-α for IgA turned out to be significant, and for IgG - the concentration of IL-12 and IL-2 had strong positive effects, whereas the concentration of IL-1 had a negative one. In responses to the mumps virus, for IgA, the level of CD4+CD45RA+CD62L- and the concentration of IL-1 turned out to be significant, and for IgG, the concentration of IL-2 had a positive effect, whereas the level of CD19+CD27+ had a negative one. In the humoral response to the rubella virus, the level of CD27+ and CD25+CD127+ cells had a negative effect on IgA, and the level of CD45RA+ had a positive effect on the level of IgG while the levels of CD3+CD8-CD38+ and CD127+ had a negative effect, as did the concentrations of IFN-γ and IL-10. Why does one get such different predictors?
On the one hand, it should be noted that the composition of the used vaccine includes live attenuated viruses, and not a pure protein antigen. This is important because live viruses are able to evade various influences of the immune system in different ways, moreover, they can themselves affect the immune system. Different viruses use different ways of such influence. Therefore, a uniformity in immune responses to different viruses should not be expected. Also, the formation of IgG and IgA antibodies may differ. Whereas IgG antibodies are predominantly formed as a result of the maturation of B cells in the germinal centers, the IgA response can also develop along the extrafollicular pathway, and there are other cells, other interactions and other cytokines involved.
On the other hand, the predictors we found do not contradict modern ideas about the functioning of the immune system. Indeed, the level of double positive CD45RA+CD45RO+ cells reflects the activation of lymphocytes, and TNF-α stimulates the immune response. Interestingly, the concentration of IL-2 had a positive effect on the level of IgG for measles and mumps viruses. This cytokine has a stimulatory effect on the immune response, and measles and mumps viruses are related viruses. The fact that the level of previously existing memory B cells (CD19+CD27+) has a negative impact on the formation of a new IgG response also does not contradict the known facts about competition between existing and newly formed memory cells. That the levels of T cells and activated T cells (CD27+ and CD25+CD127+) can negatively influence the level of the emerging IgA response to rubella is also not surprising, as is the negative impact of IL-10, IFN-γ and the level of initially activated helpers (CD3+CD8-CD38+).
With regard to cellular responses to measles and rubella viruses, it is very interesting that the level of antigen dependent lymphocyte proliferation in both cases was influenced by the level of NKT cells (CD3+CD16/56+). These cells are known to be involved in responses to non-protein antigens, but the effects of these cells are still not well understood. At the same time, the level of NK cells (CD3-CD16/56+) negatively affected the proliferative response to the measles virus. Probably, NK cells, being the first line of antiviral defense, can reduce the amount of the vaccine virus and thus reduce the proliferative response of the adaptive immune response to this virus. Interestingly, on the 7th day after vaccination, this population also continued to respond. It is quite natural that the level of CD8+CD107a+ in response to the measles virus is influenced by the level of activated cytotoxic cells (CD3+CD8+CD122+). The induction of IFN-γ production by this virus was negatively affected by the level of already activated helpers (CD3+CD8-CD122+), which is quite understandable, since if any immune response is already launched, it interferes with the formation of a new one. The level of CD8+CD107a+ responding to the rubella virus was affected by the level of CD27+ cells, this marker is present on all naive T cells. Also, CD27+ cells influenced the antigen-induced production of IFN-γ, which is not contradictory. In addition to all the above, it should be noted that the human population is very heterogeneous in terms of HLA, and this imposes restrictions on the presentation of certain antigens. This introduces additional interference into the process of modeling immune responses. Interestingly, the same immune parameters that responded on the 7th day after vaccination (CD3-CD8-CD38+HLADR+, CD3+HLADR+, CD3+CD8+CD122+, CD3+CD45R0+CD4+CD161+) turned out to be predictors for the level of the common proliferative response to PHA. These predictors were obtained by different methods, and such a coincidence, on the one hand, suggests that the response of the immune system to vaccination on the 7th day is due to the proliferative reaction of activated lymphocyte clones. On the other hand, such a coincidence indicates that the modelling-based choice of predictors is not random, but really reflects the processes under study.
Thus, it should be emphasized that the baseline immune status predictors identified by us do not contradict modern conceptions about the development of the humoral and cellular immune response to viral antigens. We hope that further research in this area will clarify the range of predictors and deepen our understanding of the interaction of various parts of the immune system in the formation of immune responses to viruses.”

Reviewer 2 Report
Major comments
- The discussion lacks details about the biological significance of the results. - No explanation of the strengths or limitations of the study is provided. Many of the immune parameters are similar in nature, but the modeling results are very different – no discussion provided on why this may be the case.
- Very little information was provided on why these 3 predictive modeling approaches were used.
- English grammar review is needed – there are some verb inconsistencies.
- Why wasn’t mumps antigen used to evaluate the mumps-specific immune responses?
Minor comments
· Lines 77-79: where is Section 1?
· Line 93: replace ‘totally’ with ‘in total’
· Line 98: replace ‘parts’ with ‘aliquots’ repeat that language in sections 2.4, 2.5, and 2.6.
· Line 109: how much virus culture extract was used? Protein quantification used to standardize? How long was the stimulation?
· Line 117: do we know the protein content of the antigen? If not, was the same stock used for all stimulations?
· Line 336: sentence doesn’t make sense, please revise
· Figure 1. What do colors represent in plots A, B, and C?
Author Response
Response to Reviewer 2 comments
We thank the Reviewer for insightful comments and the thorough work on our manuscript. All the recommendations have been addressed in the revised manuscript as described below.
Major comments
(1) - The discussion lacks details about the biological significance of the results.
Our response:
We have provided an extensive discussion of the implications concerning the immunological mechanisms underlying the identified predictors of responses to the vaccine:
“The calculated predictors for humoral and cellular responses to the three viruses included in the vaccine differ from each other. Indeed, the well-known fact that different people form humoral and cellular responses of different levels to the same dose of the vaccine is usually explained by the initial state of the immune system. But it is still not known for certain which parameters have such an effect. Having studied a wide range of parameters of the immune system before vaccination, we expected to get well-explained parameters as predictors from the point of view of modern ideas about the work of the immune system. The complexity of the modeling we have undertaken lies in the fact that the immune system functions as a network intertwined system, that is, various parameters can affect other parameters, and the effect may be pleiotropic. Having carried out mathematical modeling of the processes of formation of immune responses to vaccination, we tried to answer the question posed.
So for humoral responses to the measles virus, the level of double-positive CD45RA+CD45RO+ cells and the concentration of TNF-α for IgA turned out to be significant, and for IgG - the concentration of IL-12 and IL-2 had strong positive effects, whereas the concentration of IL-1 had a negative one. In responses to the mumps virus, for IgA, the level of CD4+CD45RA+CD62L- and the concentration of IL-1 turned out to be significant, and for IgG, the concentration of IL-2 had a positive effect, whereas the level of CD19+CD27+ had a negative one. In the humoral response to the rubella virus, the level of CD27+ and CD25+CD127+ cells had a negative effect on IgA, and the level of CD45RA+ had a positive effect on the level of IgG while the levels of CD3+CD8-CD38+ and CD127+ had a negative effect, as did the concentrations of IFN-γ and IL-10. Why does one get such different predictors?
On the one hand, it should be noted that the composition of the used vaccine includes live attenuated viruses, and not a pure protein antigen. This is important because live viruses are able to evade various influences of the immune system in different ways, moreover, they can themselves affect the immune system. Different viruses use different ways of such influence. Therefore, a uniformity in immune responses to different viruses should not be expected. Also, the formation of IgG and IgA antibodies may differ. Whereas IgG antibodies are predominantly formed as a result of the maturation of B cells in the germinal centers, the IgA response can also develop along the extrafollicular pathway, and there are other cells, other interactions and other cytokines involved.
On the other hand, the predictors we found do not contradict modern ideas about the functioning of the immune system. Indeed, the level of double positive CD45RA+CD45RO+ cells reflects the activation of lymphocytes, and TNF-α stimulates the immune response. Interestingly, the concentration of IL-2 had a positive effect on the level of IgG for measles and mumps viruses. This cytokine has a stimulatory effect on the immune response, and measles and mumps viruses are related viruses. The fact that the level of previously existing memory B cells (CD19+CD27+) has a negative impact on the formation of a new IgG response also does not contradict the known facts about competition between existing and newly formed memory cells. That the levels of T cells and activated T cells (CD27+ and CD25+CD127+) can negatively influence the level of the emerging IgA response to rubella is also not surprising, as is the negative impact of IL-10, IFN-γ and the level of initially activated helpers (CD3+CD8-CD38+).
With regard to cellular responses to measles and rubella viruses, it is very interesting that the level of antigen dependent lymphocyte proliferation in both cases was influenced by the level of NKT cells (CD3+CD16/56+). These cells are known to be involved in responses to non-protein antigens, but the effects of these cells are still not well understood. At the same time, the level of NK cells (CD3-CD16/56+) negatively affected the proliferative response to the measles virus. Probably, NK cells, being the first line of antiviral defense, can reduce the amount of the vaccine virus and thus reduce the proliferative response of the adaptive immune response to this virus. Interestingly, on the 7th day after vaccination, this population also continued to respond. It is quite natural that the level of CD8+CD107a+ in response to the measles virus is influenced by the level of activated cytotoxic cells (CD3+CD8+CD122+). The induction of IFN-γ production by this virus was negatively affected by the level of already activated helpers (CD3+CD8-CD122+), which is quite understandable, since if any immune response is already launched, it interferes with the formation of a new one. The level of CD8+CD107a+ responding to the rubella virus was affected by the level of CD27+ cells, this marker is present on all naive T cells. Also, CD27+ cells influenced the antigen-induced production of IFN-γ, which is not contradictory. In addition to all the above, it should be noted that the human population is very heterogeneous in terms of HLA, and this imposes restrictions on the presentation of certain antigens. This introduces additional interference into the process of modeling immune responses. Interestingly, the same immune parameters that responded on the 7th day after vaccination (CD3-CD8-CD38+HLADR+, CD3+HLADR+, CD3+CD8+CD122+, CD3+CD45R0+CD4+CD161+) turned out to be predictors for the level of the common proliferative response to PHA. These predictors were obtained by different methods, and such a coincidence, on the one hand, suggests that the response of the immune system to vaccination on the 7th day is due to the proliferative reaction of activated lymphocyte clones. On the other hand, such a coincidence indicates that the modelling-based choice of predictors is not random, but really reflects the processes under study.
Thus, it should be emphasized that the baseline immune status predictors identified by us do not contradict modern conceptions about the development of the humoral and cellular immune response to viral antigens. We hope that further research in this area will clarify the range of predictors and deepen our understanding of the interaction of various parts of the immune system in the formation of immune responses to viruses.”
(2) - No explanation of the strengths or limitations of the study is provided. Many of the immune parameters are similar in nature, but the modeling results are very different – no discussion provided on why this may be the case.
Our response:
The limitations of the overall approach are highlighted in the consideration of immunological aspects of the findings presented above. We have also added an extensive discussion of essential mathematical aspects of the study to the Discussion section as described below:
“To identify baseline predictors of post-vaccination immune response, we utilized various screening strategies which are known as feature selection methods in machine learning literature [1]. The feature selection methods are used to construct competitive predictive models which are then compared to select the final optimal one. The correlation-based (screening strategy (a)) and regularization-based (implicit strategy (c)) methods do not imply assumptions about the nature of connection between predictive and response variables. In contrast, supervised learning strategy (b) assumes that baseline variables that either change (b1) or do not change (b2) coherently on day 7 among individuals are predictive of the immune response. This strategy produced the best final model for unspecific and measles-specific cell proliferation levels. The unspecific response is predicted with parameters that changed coherently across individuals while the measles-specific cell proliferation levels are explained with variables which change individually in different directions during the first 7 days post-vaccination.
We note that the results of predictive modeling should be regarded as part of explanatory analysis to select potential baseline predictors among a large subset of immune status parameters. Due to a limited number of observations for some baseline variables, they were excluded from being candidates for predictive modeling although some of them (TGF-β and IL-17A) correlate with response variables strongly (TGF-β significantly and positively correlates with measles-specific IgG, IL-17A – with measles-specific IgA). More research is needed to validate the results and refine the baseline predictors and their mechanistic links with various aspects of post-vaccination immune response.”
[1] Kuhn, M.; Johnson, K. Applied Predictive Modeling; Springer New York: New York, NY, 2013; ISBN 9781461468486.
(3) - Very little information was provided on why these 3 predictive modeling approaches were used.
Our response:
We have added a broader comment on the essential aspects of the mathematical modelling approaches of the study to the Discussion section as described above. It states, that “Using various modelling approaches is required to ensure an overall robustness of the results”.
(4) - English grammar review is needed – there are some verb inconsistencies.
Our response:
The document has been additionally checked for the wording and grammatical errors.
(5) - Why wasn’t mumps antigen used to evaluate the mumps-specific immune responses?
Our response:
The mumps antigen was not used because it was not available.
Minor comments
(6) - Lines 77-79: where is Section 1?
Our response:
According to the journal template, Introduction is refereed as Section 1.
(7) - Line 93: replace ‘totally’ with ‘in total’
Our response:
Corrected.
(8) - Line 98: replace ‘parts’ with ‘aliquots’ repeat that language in sections 2.4, 2.5, and 2.6.
Our response:
Corrected.
(9) - Line 109: how much virus culture extract was used? Protein quantification used to standardize? How long was the stimulation?
Our response:
We have added the following details:
“As antigens, a twin-ether extract of measles and rubella virus culture prepared by the E. Norrby method was used. Antigens were preliminarily titrated on the mononuclear cells of several donors so that the selected concentration induced cellular responses in donors vaccinated and recovered from measles or rubella and did not cause such responses in donors who had not previously been in contact with these viruses.”
Norrby E. Separation of measles virus components by equilibrium centrifugation in CsCl gradients. I. crude and tween and ether treated concentrated tissue culture material Arch Gesamte Virusforsch. 1964;14:306-18. doi: 10.1007/BF01555823.
(10) - Line 117: do we know the protein content of the antigen? If not, was the same stock used for all stimulations?
Our response:
As the protein content was not known, we have added: “The same stock was used for all stimulations.”
(11) - Line 336: sentence doesn’t make sense, please revise
Our response:
We have revised the sentence to read:
“Prediction of the vaccine-induced immune responses using immune signatures measured at baseline prior to vaccination is a problem of fundamental and medical importance.
(12) - Figure 1. What do colors represent in plots A, B, and C?
Our response:
In Plot A, the color refers to data availability. In plots B, C and D different colors indicate different individuals (the respective comment has been added to Figure legend). In Plot E, the missing data are marked in red.
